# Ramadan Fasting Improves Body Composition without Exacerbating Depression in Males with Diagnosed Major Depressive Disorders

**DOI:** 10.3390/nu13082718

**Published:** 2021-08-07

**Authors:** Haitham Jahrami, Ahmed S. BaHammam, Eman Ahmed Haji, Nicola L. Bragazzi, Ihab Rakha, Amani Alsabbagh, Boya Nugraha, Stefan M. Pasiakos

**Affiliations:** 1Ministry of Health, Manama 410, Bahrain; EHaji@health.gov.bh (E.A.H.); IRakha@health.gov.bh (I.R.); ASabbagh4@health.gov.bh (A.A.); 2Department of Psychiatry, College of Medicine and Medical Sciences, Arabian Gulf University, Manama 323, Bahrain; 3University Sleep Disorders Center, Department of Medicine, College of Medicine, King Saud University, P.O. Box 225503, Riyadh 11324, Saudi Arabia; ashammam2@gmail.com; 4The Strategic Technologies Program of the National Plan for Sciences and Technology and Innovation in the Kingdom of Saudi Arabia, Riyadh 11324, Saudi Arabia; 5Laboratory for Industrial and Applied Mathematics, Departments and Statistics, York University, Toronto, ON M3J 1P3, Canada; 6Department of Rehabilitation Medicine, Hannover Medical School, 30625 Hannover, Germany; boya.nugraha@gmail.com; 7Military Nutrition Division, U.S. Army Research Institute of Environmental Medicine, Natick, MA 01760, USA; stefan.m.pasiakos.civ@mail.mil

**Keywords:** depression, intermittent fasting, mood disorders, metabolic syndrome

## Abstract

Background: Ramadan fasting (RF) is a form of intermittent fasting that generally improves body composition and related metabolic profiles. Whether RF exacerbates depressive symptomatology in individuals diagnosed with major depressive disorder (MDD) is undetermined. Methods: 100 men, who lived in Bahrain and were between the ages of 18 and 64 years with an established diagnosis of MDD, participated in this 4-week study. Based on preference, participants were assigned to a fasting group (FG, *n* = 50) and a non-fasting group (NFG, *n* = 50). The FG engaged in fasting from 03:40 to 18:10 (dawn and dusk timings). Changes in depressive symptoms, body mass, body composition, and components of metabolic syndrome were measured. Results: There were no significant changes in depressive symptoms within the FG vs. NFG after controlling for baseline covariates: mean difference 0.49 (SE = 0.63), *p* = 0.43. No adverse effects were reported in either group. The FG experienced significant reductions in body mass, 1.87 kg, *p* = 0.001; body mass index, 0.69 kg/m^2^, *p* = 0.001; body fat, 0.87%, *p* = 0.001; body surface area, 0.03 m^2^, *p* = 0.001; and lean mass, 0.77 kg, *p* = 0.001. Conclusions: RF did not negatively affect depressive symptoms and improved body composition, suggesting short-term intermittent fasting may be a safe dietary practice for adult males with MDD.

## 1. Introduction

Ramadan fasting (RF) is the religious fast practiced by 1.8 billion Muslims during the month of Ramadan (the ninth month of the Islamic calendar year) [1], and involves abstinence from drink, food, coitus, and smoke from dawn to dusk [2]. RF is considered a form of intermittent fasting and entails a shift to exclusively nocturnal feeding [3]. RF is a departure from conventional routines in many aspects, including changes in major meal timing, sleep, appetite, energy metabolism, and hormonal response to food [4]. There are several reasons for not fasting during the month of Ramadan in adulthood, including people with illnesses for whom fasting may exacerbate their condition [5]. 

Major depressive disorder (MDD) is the most common neuropsychiatric disorder, which affects more than 300 million individuals or about 4.5% of the global population [6]. Navigating Ramadan with mental illness, particularly MDD, can be challenging. Time-restricted feeding [7], changes in appetite [8], staying up late until the early hours of the morning for suhoor (the predawn meal), and changes in social interaction [8] may all interfere with symptoms of depression [9]. Furthermore, patients frequently reported that they feel demotivated, socially isolated, and guilty because they are unable to experience enjoyment with others [10,11].

The effects of RF on physical health are well characterized, including a reduction in body mass [12] and improvements in the various components of metabolic syndrome (MS), such as waist circumference (WC), systolic blood pressure (SBP), triglycerides (TG), high-density lipoprotein (HDL) cholesterol [13], and fasting blood glucose (FBG) [14]. Most of the previous studies were performed on healthy subjects, and only a few studies involved patients with psychiatric illnesses [2,15]. The prevalence of obesity, MS, and related metabolic abnormalities in patients with MDD is extremely high [16,17], and weight loss is often a challenge [18,19]. In terms of clinical practice, it is not clear if MDD patients can fast and experience the same positive effects on body composition and related MS markers as those without MDD, especially without exacerbating MDD symptomatology.

To date, the effects of RF on depressive symptoms, weight loss, MS-related abnormalities, and body composition in patients with MDD are not well described. This makes giving medical advice to patients with MDD on fasting a complicated and challenging task. Hence, this study aimed to investigate the effects of RF on the mental and physical health of patients with MDD. We hypothesized that RF will exacerbate depressive symptoms but will provide favorable outcomes to physical health, including body composition and related MS markers.

## 2. Materials and Methods

### 2.1. Research Design

The current research employed a two-group non-randomized quasi-experimental controlled design and was conducted in the month of Ramadan (1441 Hijri), which was between 23 April 2020 and 23 May 2020. To support the organizing and reporting, the Transparent Reporting of Evaluations with Non-randomized Designs (TREND) statement was utilized [20].

### 2.2. Ethical Issues

The research protocol was reviewed and approved by the research ethics committee in secondary health care, Ministry of Health, Bahrain (approval code MOH/SHCREC/EF0215). Written informed consent was requested and acquired from each participant before registration in the study. All procedures performed in this study followed the ethical standards of the Ministry of Health research committee and the 1964 Helsinki Declaration and its later amendments or comparable ethical standards. 

### 2.3. Setting, Participants, Group Assignment, and Sample Size Calculation

Outpatient participants were recruited from the outpatient clinic, Psychiatric Hospital, Ministry of Health, Bahrain. The Psychiatric Hospital is the national and only public center for diagnosis, treatment, and follow-up of cases with serious mental illnesses in the Kingdom of Bahrain.

This study involved male patients diagnosed with MDD according to the clinical descriptions and diagnostic guidelines of the classification of mental and behavioral disorders chapter within the international classification of disease version 10 (ICD-10) [21]. Female patients were not included because Muslim women are exempted from RF during the period of menses, thus making fasting days interrupted. Other inclusion criteria were adults aged between 20 and 64 years of age, competent to provide informed consent, and willing to participate. We excluded patients with a medical excuse from fasting, those who were dieting, those following a lifestyle management program, or those involved in other interventional studies.

Participants were assigned to the study groups using a non-randomized approach. Before enrollment to groups, patients were asked if they were planning “intentions” to fast for Ramadan by the principal investigator, and their answer determined assignment to the fasting group (FG) vs. non-fasting group (NFG). Group assignment was further validated before data collection based on participants’ initial report. The FG served as the intervention, and the NFG was considered as the control. Figure 1 demonstrates a flow diagram of participants’ recruitment. 

Using sample size calculations for non-randomized studies, we projected that 35 participants are needed per group to provide 80% power to reveal a significant difference of one kilogram in body mass difference between groups using an independent samples *t*-test with an alpha of 0.05. The anticipated change in body mass of approximately one kilogram (an effect size of 0.20) after fasting is based on findings of a recent systematic review and meta-analysis [12]. To account for any potential loss of follow-up during the study, we assumed a 30% dropout rate, and this aimed to include 50 patients per group at the start of the study.

### 2.4. Measurements/Endpoints

Data were collected from each participant at baseline (T1 = one week before the month of Ramadan) and after fasting (T2 = after completing 28 days of the fasting month). The daily fasting time was approximately 870 min from 03:40 to 18:10 (14 h and 30 min) per day according to Umm Al-Qura calculation methods of Muslim prayers for Manama, Kingdom of Bahrain. No specific dietary, physical activities or other lifestyle modification suggestions were given during the study period.

All measurements were facilitated by a single-blinded research assistant. Data collection forms included the following items: sociodemographic variables (at T1 only), an assessment of physical activity (PA) (at T1 only), assessment of depressive symptoms using the Patient Health Questionnaire-9 (PHQ-9) (at T1, T2), anthropometric and direct-segmental, multi-frequency bioelectrical impedance analysis (BIA) (at T1, T2), blood pressure (at T1, T2), lipid profile (at T1, T2), and finally fasting blood sugar (FBS) (at T1, T2).

Sociodemographic variables and assessment of PA included age, sex, marital status, employment status, smoking status, and level of physical activity engagement. PA was measured using the validated outcome measure International Physical Activity Questionnaire—Short Form (IPAQ-SF) [22]. The IPAQ-SF evaluates the frequency and duration of light-, moderate- and vigorous-intensity PA. The total metabolic equivalent task (MET) per minute/week was computed accordingly, and patients were considered to be minimally active if they met 600 MET min/week, and individuals with 3000 MET minutes/week are considered to be Health-Enhancing Physical Activity (HEPA) active [22]. The Arabic language validated short version was used (http://www.ipaq.ki.se, accessed on 6 January 2020).

The PHQ-9 [23] was used to measure self-reported depressive symptoms. The PHQ-9 has a sensitivity of 88% and a specificity of 88% for major depression disorders at 10 points [23]. The Arabic language validated version of the PHQ-9 [24] was used. The severity of depressive symptoms was determined using the PHQ-9 using the following scores: 0–4 no depressive symptoms, 5–9 mild depression, 10–14 moderate depression, 15–19 moderately severe depression, 20–27 severe depression [23].

Anthropometric and BIA included body mass (kg), height (cm), waist circumference (cm), hip circumference (cm), lean mass (kg), fat mass (kg), total body water percent (%), and percent body fat (%). Body mass index (BMI) was estimated in kg/m^2^ by dividing the weight (kg) by the squared height (m) for each person [25]. The waist–hip ratio (WHR) was estimated by dividing the waist circumference (cm) by the hip circumference (cm). The waist circumference was defined as the total distance of the lower margin of the last palpable ribs and the top of the iliac crest, using a stretch-resistant tape. Hip circumference is the total distance around the widest portion of the buttocks, with the tape parallel to the floor [26]. BIA was performed using InBody 770 (InBody, CA, USA).

Before completing the BIA measurements, all accessories, metals, and/or jewelry were removed following the manufacturer’s recommendations, and each participant was asked to eliminate extra body fluids by urine. The influence of hydration and physical exertion on BIA measures was reduced since all individuals fasted for eight to ten hours before each of the two time periods, decreasing intra-individual variability. Furthermore, BIA and all other parameters were collected at the same time of day for T1 and T2.

After ten minutes of rest, blood pressure, oxygen saturation, and heart rate were monitored using a digital blood pressure monitor (Dräger, Vista 120, Lübeck, Germany), with individuals in a sitting position. Mean arterial pressure (MAP) was computed using the following formula: ((SBP) + (2 × DBP))/3 [27].

After assessing blood pressure, samples of venous blood (10 mL) were taken. To reduce the influence of time on the assessed variables, venipuncture collection was performed after eight to ten hours of fasting at both time periods. Blood lipid profile and FBS were measured. The lipid profile included total cholesterol (TC), high-density lipoprotein cholesterol (HDL), low-density lipoprotein cholesterol (LDL), and triglycerides (TG). The blood lipid profile and FBS were analyzed using Simens Dimension EXL 200 and Simens ADIVA Chemistry XPT (Simens, Munich, Germany).

The MAP was essential to calculate the metabolic syndrome z-score based on the methodology proposed by the Division of Cardiology at Duke University [28]. However, in the current research, we used the new world-wide definition of metabolic syndrome presented by the International Diabetes Federation (IDF) [29,30]. The z-score was calculated for each patient using his data, the IDF criteria, and the standard deviation of the entire data. The equation used was: z-score = [((1.03 − HDL)/0.72) + ((TG − 1.7)/0.51) + ((FBG − 5.6)/1.1) + ((WC − 94)/10.56) + ((MAP − 95)/10.9)]. Negative z-scores (and negative z-score changes) are indicative of lower risk [28]. Appendix A provides a summary of the study parameters and an interpretation key.

### 2.5. Statistical Analyses

Before the start of data analyses, the data were visualized using histograms and box plots to check normality and identify any potential outliers. The Shapiro–Wilk test was deployed to formally test the normality of the study variables. Descriptive statistics were used to summarize the data and involved mean and standard deviation (for continuous variables) and absolute and relative frequencies (for categorical data).

Paired samples *t*-test was used to compare the difference in the study variables before and after fasting. The independent samples *t*-test was used to compare the change (before–after) between participants in the fasting group vs. participants in the NFG. The effect size was estimated via Cohen’s d and was interpreted as follows: 0.20 as small, 0.50 as moderate, ≥0.80 as large [31]. For a comparison of two means, Cohen’s d is an acceptable effect size. It can be used to complement the reporting of *t*-test and ANOVA results [31]. Cohen’s d represented the standardized mean difference between two groups by subtracting one group’s mean from the other’s (M1—M2) and dividing the result by the population’s standard deviation (SD) [31].

Analysis of covariance (ANCOVA) was used as a general linear model approach to examine the magnitude of change within FG vs. NFG in the significant study parameters covarying for marital status, employment status, smoking status, and psychotropic medication (as factors) and age, physical activity, baseline body mass, and baseline PHQ-9. Post hoc comparisons are based on estimated marginal means using Tukey correction. We used two-tailed tests, and results with *p* values less than 0.05 were considered statistically significant. All statistical analyses were performed using R for statistical computing 4.0.3 [32].

## 3. Results

One hundred male patients with MDD were involved in this study. The mean age was 46 ± 10 years of age, and about 50% were married, 60% unemployed, 35% current smokers, and more than 60% were physically inactive with an average of 740 MET min/week. All patients were on pharmacological treatment for depression at the time of the study. The antidepressants used were: selective serotonin reuptake inhibitors (SSRIs) (51%), serotonin–norepinephrine reuptake inhibitors (SNRIs) (32%), tricyclic antidepressants (TCAs) (11%), and atypical antidepressants (6%). The fasting and non-fasting groups were similar in their characteristics, suggesting that the two groups are comparable. Table 1 provides the baseline socio-demographics of the study participants. At baseline, the majority of our patients were in the moderately severe depression group (PHQ-9 ≥ 14); specifically, 88% and 86% were in that category for the FG and NFG, respectively.

The FG had statistically significant changes in nine parameters, including PHQ-9, body mass, BMI, BSA, %BF, LM, HDL, TG, and WC, after four weeks of fasting. The NFG had statistically significant changes in seven parameters, including PHQ-9, body mass, BMI, BSA, %BF, TBWP, LM, and WC, after four weeks of follow-up (see Table 2).

Fasting produced a statistically significant reduction in PHQ-9, body mass, BMI, BSA, BFP, and HDL. Nonetheless, fasting also resulted in a significant loss of LM. Improvements in central obesity and HDL did not alter the MS z-score in the FG compared to the NFG *p* = 0.85 (see Table 3).

Results from the ANCOVA suggest that fasting did not affect depressive symptoms—the mean difference of PHQ-9 was 0.49 (SE = 0.63), *p* = 0.436, Cohen ES = 0.17. However, fasting produced a large change in the following: body mass 1.86 kg (SE = 0.06), *p* = 0.001, Cohen ES = 6.35; BMI 0.69 kg/m^2^ (SE = 0.03), *p* = 0.001, Cohen ES 5.23; BFP 0.91% (SE = 0.09), *p* = 0.001, Cohen ES = 2.09; BSA 0.03 m^2^ (SE = 0.01), *p* = 0.001, Cohen ES 0.81; LM 0.79 kg (SE = 0.10), *p* = 0.001, Cohen ES = 1.63 (see Table 4).

## 4. Discussion

Four weeks of intermittent fasting during the month of Ramadan performed by male Muslim patients with depression did not exacerbate depressive symptoms. The fasting produced a significant loss in body mass and body fat. RF appears to be safe for psychiatric symptoms and apparently beneficial for physical health for those individuals with MDD. Clinicians need to provide advice to their patients on RF and other forms of intermittent fasting after a thorough assessment of their patient’s psychiatric and medical condition. Frequent follow-ups as continued care with patients with MDD during Ramadan fasting are encouraged.

Our data showed that no significant changes in self-reported depression scores were observed after RF. A limited number of studies have assessed the effect of RF on depressive symptoms in patients with psychiatric conditions and reported a marginal improvement in mood and emotional response [10]. On the other hand, a few studies assessed the impact of experimental or RF on mood and reported conflicting results [9,10,33]. These discrepancies in the reported results could be related to the assessment method used to evaluate the changes in depressive symptoms or severity. Previous studies relied mainly on patients’ reports or clinical interviews and lacked the use of a reliable and valid outcome measure of depressive symptoms [9,11]. The deterioration in depressive symptoms observed in the non-fasting group can possibly be explained by the fact that previous studies in healthy people [34] and patients with diabetes [35] showed that fasting can lead to an improvement in stress scores, which may explain our findings.

Additionally, data extrapolated from clinical and pre-clinical caloric restriction (CR) studies indicated that short-term CR protocols might induce an antidepressant effect [7]. Several mechanisms have been proposed to explain the antidepressant effects of CR, including orexin signaling activation, increased CREB (cAMP response element-binding) phosphorylation, and the neurotrophic effects of the release of endorphins and ketone production [36,37,38].

Body mass loss and body fat reduction after RF were previously reported with a similar magnitude in two systematic reviews and meta-analyses of healthy individuals [12,39]. Weight loss and loss of fat-free mass and body fat percentage after RF were observed at higher magnitudes in overweight or obese individuals [39]. Our participants were mostly overweight or obese at baseline, and this may explain why they experienced the benefit of fasting at a pronounced magnitude.

Another interesting finding of the current study is that body mass loss did not reduce the overall risk for MS and related metabolic markers. Previous research showed that patients with depression had an increased risk of MS compared to age- and sex-matched controls [40], and lifestyle interventions are needed. Powered, controlled, longitudinal studies demonstrating the long-term benefits of RF (or other forms of intermittent fasting) on body mass and its impact on MS and related metabolic abnormalities in clinically depressed individuals are encouraged.

Another finding by our research is that almost half of the weight loss was in lean mass. This is in disagreement with RF in healthy participants fasting for approximately 17 h per day [41]. However, our findings are consistent with the findings of a recent randomized clinical controlled trial of the 16:8 diet (proxy to Ramadan fast), and suggest that time-restricted feeding and intermittent fasting cause a significant loss in lean mass in overweight and obese individuals [42]. This is a source of concern because muscle loss may affect power, mobility, and metabolic health [43,44]. Monitoring physical activity thresholds and protein intake during future controlled trials is needed to examine the covariate nature of these two factors on lean mass, as previous studies have reported changes in dietary contents during Ramadan [45,46].

The current paper has strengths and limitations. The major strength includes being the first study to investigate the effects of RF on depressive symptoms, weight loss, MS and related metabolic abnormalities, and body composition in patients with MDD using multifaceted methodologies that assessed both mental and physical health. As a non-randomized approach was used, the TREND statement was deployed to mitigate concerns with bias. Future randomized studies are needed to eliminate the bias of the self-selection of fasting as an intervention. Randomization is truly unfeasible in studies investigating religious types of fasting [47]; nonetheless, more rigorous research is needed.

The limitations include the lack of objective measurement of physical activity and sleep duration and timing as potential covariates for the study parameters. Thus, future studies need to include the objective measurement of physical activities and sleep using actigraphy or wearable devices, and they need to record the food intake and control for them in the analyses. Another limitation is the lack of follow-up measurements after RF to better understand the nature of changes. Such long-term follow-up studies need to also account for the daily estimated energy intake, resting energy expenditure, and total energy expenditure.

## 5. Conclusions

Four weeks of intermittent fasting during the month of Ramadan performed by Muslim patients with depression did not exacerbate depressive symptoms. RF was associated with a significant loss in body mass and body fat, suggesting that RF is potentially safe and likely beneficial for males with MDD. Further research is needed to confirm the safety and efficacy of fasting for people with depression.

## Figures and Tables

**Figure 1 nutrients-13-02718-f001:**
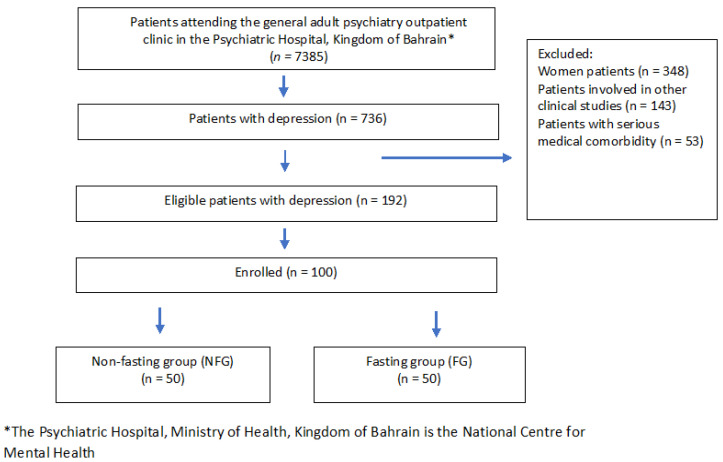
Flow diagram of participants’ recruitment.

**Table 1 nutrients-13-02718-t001:** Socio-demographics of the study participants.

Variable *	All Participants *n* = 100	Fasting Group *n* = 50	Non-Fasting Group *n* = 50	*p* Value **
Age (years)	45.96 ± 10.33	43.38 ± 10.06	46.54 ± 8.35	0.1
Weekly Physical Activity (MET/Week)	739.07 ± 1193.81	643.68 ± 1082.28	834.46 ± 1299.84	0.5
Marital Status				0.3
-Married	51 (51%)	23 (46%)	28 (56%)	
-Single	49 (49%)	27 (54%)	22 (44%)	
Sex				-
-Male	100 (100%)	50 (100%)	50 (100%)	
Employment Status				0.8
-Employed	43 (43%)	22 (44%)	21 (42%)	
-Unemployed	57 (57%)	28 (56%)	29 (58%)	
Smoking Status				0.5
-Current smoker	35 (35%)	16 (32%)	19 (38%)	
Weekly Physical Activity Levels				0.9
-HEPA active	9 (9%)	4 (8%)	5 (10%)	
-Inactive	63 (63%)	32 (64%)	31 (62%)	
-Minimally active	28 (28%)	14 (28%)	14 (28%)	
Severity of depression at baseline (PHQ-9) ****				0.2
-No depression	Nil (0%)	Nil (0%)	Nil (0%)	
-Mild depression	1 (1%)	Nil (0%)	1 (2%)	
-Moderate depression	12 (12%)	6 (12%)	6 (12%)	
-Moderately severe depression	67 (67%)	30 (60%)	37 (74%)	
-Severe depression	20 (20%)	14 (28%)	6 (12%)	

* Frequency and percentage OR arithmetic mean and standard deviation; ** Independent Samples *t*-test; **** PHQ-9 0–4 no depressive symptoms, 5–9 mild depression, 10–14 moderate depression, 15–19 moderately severe depression, 20–27 severe depression.

**Table 2 nutrients-13-02718-t002:** Results of study parameters before and after fasting by group (fasting group vs. non-fasting group).

Variable * (Unit)	Fasting Group				Non-Fasting Group			
	Pre	Post	*p* Value **	Effect Size	Pre	Post	*p* Value *	Effect Size
Body mass (kg)	82.04 ± 12.19	80.55 ± 12.25	0.001 ***	4.778	86.19 ± 12.27	86.60 ± 12.27	0.001 ***	−1.400
Height (cm)	168.43 ± 13.31	168.43 ± 13.31	-	-	168.64 ± 14.59	168.64 ± 14.59	-	-
BMI (kg/m^2^)	29.47 ± 6.45	28.93 ± 6.39	0.001 ***	3.728	30.98 ± 6.53	31.13 ± 6.56	0.001 ***	−1.251
%BFP (%)	29.48 ± 8.56	28.74 ± 8.42	0.001 ***	1.520	32.20 ± 8.93	32.34 ± 8.90	0.010 ***	−0.399
%TBWP (%)	39.40 ± 11.64	41.14 ± 4.38	0.300	−0.138	36.00 ± 11.48	41.20 ± 3.98	0.010 ***	−0.397
BSA (m^2^)	1.95 ± 0.18	1.92 ± 0.18	0.001 ***	0.587	2.01 ± 0.17	2.01 ± 0.17	0.300	−0.141
LM (kg)	54.30 ± 6.60	53.78 ± 6.65	0.001 ***	1.030	55.98 ± 6.18	56.22 ± 6.20	0.003 ***	−0.556
FM (kg)	29.98 ± 9.31	30.92 ± 2.88	0.500	−0.094	30.96 ± 8.63	31.18 ± 2.87	0.900	−0.023
SBP (mmHg)	136.82 ± 21.03	135.62 ± 19.22	0.300	0.159	134.82 ± 24.82	135.12 ± 24.61	0.900	−0.008
DBP (mmHg)	76.06 ± 11.05	76.26 ± 10.97	0.900	−0.009	74.38 ± 11.63	74.52 ± 11.72	0.900	0.998
LDL (mmol/L)	1.30 ± 0.28	1.30 ± 0.30	0.900	−0.014	1.31 ± 0.24	1.35 ± 0.33	0.200	−0.199
HDL (mmol/L)	2.96 ± 0.70	3.01 ± 0.76	0.010 ***	−0.371	3.13 ± 0.74	3.12 ± 0.69	0.600	0.082
TG (mmol/L)	1.67 ± 0.56	1.56 ± 0.58	0.030 ***	−0.317	1.50 ± 0.43	1.55 ± 0.46	0.06	−0.275
TC (mmol/L)	4.88 ± 0.92	5.09 ± 0.92	0.300	−0.164	4.88 ± 0.90	5.03 ± 0.86	0.500	−0.107
FBG (mmol/L)	5.60 ± 1.17	5.73 ± 0.97	0.500	−0.095	5.72 ± 1.05	5.80 ± 0.85	0.700	−0.063
WC (cm)	93.20 ± 10.21	92.62 ± 10.26	0.010 ***	0.380	96.26 ± 10.79	95.80 ± 11.32	0.040 ***	0.298
HC (cm)	100.76 ± 11.05	100.76 ± 11.05	-	-	100.84 ± 11.13	100.84 ± 11.13	-	-
WHR (ratio)	0.93 ± 0.16	0.93 ± 0.17	0.700	0.047	0.96 ± 0.16	0.96 ± 0.16	0.600	0.077
MS z-score	−2.67 ± 2.00	−2.58, 2.28	0.635	−0.06	−3.01, 1.92	−2.86, 2.23	0.537	−0.087
PHQ-9	18.06 ± 2.85	17.76 ± 2.83	0.003 ***	−0.439	16.64 ± 2.60	18.26 ± 3.04	0.001 ***	0.740

* Arithmetic mean and standard deviation; ** Paired Samples *t*-test; *** Significant at 0.05. *p* value is for the difference between pre and post values in the fasting group and separately in the non-fasting group. BMI = body mass index; %BFP = body fat percentage; %TBWP = total body water percentage; BSA = body surface area; LM = lean mass; FM = fat mass; SBP = systolic blood pressure; DBP = diastolic blood pressure; LDL = low-density lipoprotein cholesterol; HDL = low-density lipoprotein cholesterol; TG = triglycerides; TC = total cholesterol; FBG = fasting blood glucose; WC = waist circumference; HC = hip circumference; WHR = waist to hip ratio; MS z-score = metabolic syndrome z-score; PHQ-9 = patient health questionnaire-9.

**Table 3 nutrients-13-02718-t003:** Independent samples *t*-test comparing the magnitude of change within fasting group vs. non-fasting group in the study parameters.

Variable * (Unit)	Δ Fasting Group	Δ Non-Fasting Group	*p* Value **	Effect Size
Body mass (kg)	1.49 ± 0.31	−0.41 ± 0.29	0.001 ***	6.28
Height (cm)	0.00 ± 0.00	0.00 ± 0.00	NA	NA
BMI (kg/m^2^)	0.54 ± 0.15	−0.15 ± 0.12	0.001 ***	5.20
%BFP (%)	0.74 ± 0.49	−0.14 ± 0.35	0.001 ***	2.07
%TBWP (%)	−1.74 ± 12.58	−5.20 ± 13.10	0.18	0.27
BSA (m^2^)	0.03 ± 0.04	0.00 ± 0.01	0.001 ***	0.85
LM (kg)	0.52 ± 0.51	−0.24 ± 0.43	0.001 ***	1.62
FM (kg)	−0.94 ± 10.03	−0.22 ± 9.48	0.71	−0.07
SBP (mmHg)	1.20 ± 7.57	−0.30 ± 36.63	0.77	0.06
DBP (mmHg)	−0.20 ± 1.21	−0.14 ± 16.07	0.97	−0.01
LDL (mmol/L)	0.00 ± 0.16	−0.03 ± 0.17	0.33	0.19
HDL (mmol/L)	−0.05 ± 0.14	0.01 ± 0.18	0.04 ***	−0.42
TG (mmol/L)	−0.06 ± 0.19	−0.04 ± 0.16	0.66	−0.29
TC (mmol/L)	−0.21 ± 1.28	−0.14 ± 1.32	0.79	−0.05
FBG (mmol/L)	−0.13 ± 1.41	−0.07 ± 1.14	0.81	−0.05
WC (cm)	0.58 ± 1.53	0.46 ± 1.54	0.70	0.08
HC (cm)	NA	NA	NA	NA
WHR (ratio)	0.01 ± 0.03	0.001 ± 0.04	0.60	0.11
MS z-score	−0.09 ± 1.34	−0.15 ± 1.72	0.85	−0.07
PHQ-9	0.30 ± 3.88	−1.62 ± 3.69	0.01 ***	0.51

* Arithmetic mean and standard deviation; ** Independent Samples *t*-test; *** Significant at 0.05. Δ= Pretest–Posttest. BMI = body mass index; %BFP = body fat percentage; %TBWP = total body water percentage; BSA = body surface area; LM = lean mass; FM = fat mass; SBP = systolic blood pressure; DBP = diastolic blood pressure; LDL = low-density lipoprotein cholesterol; HDL = low-density lipoprotein cholesterol; TG = triglycerides; TC = total cholesterol; FBG = fasting blood glucose; WC = waist circumference; HC = hip circumference; WHR = waist to hip ratio; MS z-score = metabolic syndrome z-score; PHQ-9 = patient health questionnaire-9.

**Table 4 nutrients-13-02718-t004:** Analysis of covariance (ANCOVA) comparing the magnitude of change within fasting group vs. non-fasting group in the study parameters.

Variable	Mean Difference	SE	df	t	*p* Tukey	Cohen’s d
Body mass (kg)	1.86	0.06	88	29.68	0.001 *	6.35
BMI (kg/m^2^)	0.69	0.03	88	25.32	0.001 *	5.42
%BFP (%)	0.91	0.09	88	9.78	0.001 *	2.09
BSA (m^2^)	0.03	0.01	88	4.33	0.001 *	0.93
LM (kg)	0.79	0.10	88	7.91	0.001 *	1.63
HDL (mmol/L)	−0.06	0.03	88	−1.78	0.079	−0.38
PHQ-9	0.49	0.63	88	0.78	0.436	0.17

Note: ANCOVA adjusting for factors marital status, employment status, smoking status, and psychotropic medication and covariates age, physical activity, baseline body mass, and baseline PHQ-9. Comparisons are based on estimated marginal means, correction using Tukey. * Significant at 0.05. BMI = body mass index; %BFP = body fat percentage; BSA = body surface area; LM = lean mass; HDL = low-density lipoprotein cholesterol; PHQ-9 = patient health questionnaire-9.

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
