# Peer review of "Ramadan Fasting Improves Body Composition without Exacerbating Depression in Males with Diagnosed Major Depressive Disorders"

_nutrients, 2021, doi:10.3390/nu13082718_

Round 1

Reviewer 1 Report

The manuscript, “Ramadan fasting improves body composition without exacerbating depression in males with diagnosed major depressive disorders” is of interest to many wanting to fast, especially for religious purposes, but unsure if it would negatively affect their depressive symptoms.  It is well written but lacks a few specific details that would be beneficial for the clarity of the reader.

  1. Abstract: Remove specific statistics (Cohens) from results as complicates the results section. Also, statistics were not shown for depressive symptoms in results which makes the results section of the abstract seem unorganized.
  2. In methods, provide the name (brand, etc) for the BIA and blood pressure machines. Also, explain the process/equipment used to measure lipid profile and FBS.
  3. In the methods, please add specific required preparation by the subject prior to lab visits, as this plays a large role in BIA accuracy. Also, were they analyzed at the same time of day for baseline and post-fast/control period?
  4. In Table 1, it would be helpful to see if there was a significant difference in depression between the two groups.
  5. Why did 50 choose to fast and 50 not choose to fast? This choice seems to be a behavior difference between the 2 groups that is a limitation in the study.  It looks like according to the PHQ-9, the fasting group initially had greater symptoms of depression than the control group.  But then the fasting group decreased whereas the control group increased.  You mentioned that they felt down when they could not participate – do you think this is the reason that the control group increased their score on the PHQ-9?  Adding explanations regarding this to the discussion would be beneficial to the reader.

Author Response

Comments and Suggestions for Authors

The manuscript, “Ramadan fasting improves body composition without exacerbating depression in males with diagnosed major depressive disorders” is of interest to many wanting to fast, especially for religious purposes, but unsure if it would negatively affect their depressive symptoms.  It is well written but lacks a few specific details that would be beneficial for the clarity of the reader.

Author response: Dear Reviewer 1, we would like to thank you for your thoughtful comments and efforts towards improving our manuscript. All the comments have been taken into account when preparing the revision. We have addressed each and every comment. Changes in the manuscript are highlighted in bold red font for convenience of tracking. Thank you for your nice comments.

===========================

  1. Abstract: Remove specific statistics (Cohens) from results as complicates the results section. Also, statistics were not shown for depressive symptoms in results which makes the results section of the abstract seem unorganized.

Author response: Cohen’s d scored out for deletion, in the abstract. The Cohen’s d statistics were retained in the main results only.

Also, we added the statistics that show mean difference was not statistically different between FG and NFG after controlling for covariates 0.49 (SE=0.63), P=0.43.

===========================

  1. In methods, provide the name (brand, etc) for the BIA and blood pressure machines. Also, explain the process/equipment used to measure lipid profile and FBS.

Author response: We added the models of BIA (InBody 770) and blood pressure equipment (Dräger Vista 120).

The blood lipids profile and FBS were analyzed using (Simens Dimension EXL 200 and Simens ADIVA Chemistry XPT, Simens, Munich, Germany). We also explained that blood sample was taken after assessing blood pressure, samples of venous blood (10 mL) were taken. To reduce the influence of time on the assessed variables, venipuncture collection was done after ten hours of fasting at both time periods.

===========================

  1. In the methods, please add specific required preparation by the subject prior to lab visits, as this plays a large role in BIA accuracy. Also, were they analyzed at the same time of day for baseline and post-fast/control period?

Author response:

Before completing the BIA measurements, all accessories, metals, and/or jewelry were removed following the manufacturer's recommendations, and each participant was asked to eliminate extra body fluids by urine. The influence of hydration and physical exertion on BIA measures was reduced since all individuals fasted for eight to ten hours before each of the two time periods, decreasing intra-individual variability. BIA and all other parameters were collected at the same time of day for T1 and T2.

===========================

  1. In Table 1, it would be helpful to see if there was a significant difference in depression between the two groups.

Author response: We added a detailed categorical examination between FG and NFG at baseline. There is no statistically significant difference between both groups at baseline P=0.2.

===========================

  1. Why did 50 choose to fast and 50 not choose to fast? This choice seems to be a behavior difference between the 2 groups that is a limitation in the study.  It looks like according to the PHQ-9, the fasting group initially had greater symptoms of depression than the control group.  But then the fasting group decreased whereas the control group increased.  You mentioned that they felt down when they could not participate – do you think this is the reason that the control group increased their score on the PHQ-9?  Adding explanations regarding this to the discussion would be beneficial to the reader.

Author response: We thank the reviewer for this very important point. We added to our limitations that future randomized studies are needed to eliminate the bias of self-selection of fasting as an intervention. We made a general comment here that future studies may not be limited to Ramadan fasting by Muslims and can be extended to other experimental fasting approaches. In the Muslim community, non-fasting is considered a sin and will yield to non-participation.

===========================

Reviewer 2 Report

Jahrami and colleagues evaluated the effect of Ramadan fasting on measures of body mass/weight, metabolic health, and cardiac risk factors in a population of subjects with major depressive disorder. The study evaluated 4 weeks of fasting in a non-randomized approach. The study provides unique data on patients with a previously-diagnosed mental health disorder.

Major comments:

1. On lines 103-105 on page 3, what about the answer that the subject provide was used by the PI to determine the assignment to fasting or non-fasting groups? If they were planning to engage in Ramadan fasting or were not, was that the final answer for determining their group in the study, or did the PI take something else into consideration?

2. It is not immediately clear what the p-values presented in Table 2 are comparing. It appears on full inspect that they evaluate the difference between pre and post values in the fasting group and separately in the non-fasting group. It would be helpful to clarify this in the table to avoid any confusion about the p-values represent.

3. What is the "Effect Size" in Table 2? Is this the mean of the differences between pre- and post- values of each individual?

4. Is Table 4 presenting the results from a multivariable model? It would be helpful to clarify in the table description whether those results are all part of a multivariable model.

5. It is not clear whether this study was registered at a clinical trials site such as clinicaltrials.gov? Although not randomized, since it involves an intervention applied to humans, the study likely should be registered.

Author Response

Comments and Suggestions for Authors

Jahrami and colleagues evaluated the effect of Ramadan fasting on measures of body mass/weight, metabolic health, and cardiac risk factors in a population of subjects with major depressive disorder. The study evaluated 4 weeks of fasting in a non-randomized approach. The study provides unique data on patients with a previously-diagnosed mental health disorder.

Author response: Dear Reviewer 2, we would like to thank you for your thoughtful comments and efforts towards improving our manuscript. All the comments have been taken into account when preparing the revision. We have addressed each and every comment. Changes in the manuscript are highlighted in bold red font for the convenience of tracking. Thank you for your nice comments.

===========================

Major comments:

  1. On lines 103-105 on page 3, what about the answer that the subject provide was used by the PI to determine the assignment to fasting or non-fasting groups? If they were planning to engage in Ramadan fasting or were not, was that the final answer for determining their group in the study, or did the PI take something else into consideration?

Author response: Dear Reviewer 2, we thank you for your comment. The reviewer is absolutely correct. We asked participants if they will fast or not, and based on that we classified them into two groups. In Muslims, this is called ‘neya’ or intentions.

We further clarified that group assignment was further validated before data collection based on the participants' initial report. Just in case patients changed their opinion.

===========================

  1. It is not immediately clear what the p-values presented in Table 2 are comparing. It appears on full inspect that they evaluate the difference between pre and post values in the fasting group and separately in the non-fasting group. It would be helpful to clarify this in the table to avoid any confusion about the p-values represent.

Author response: We added a clear footnote to state that: P-value is for the difference between pre and post values in the fasting group and separately in the non-fasting group.

===========================

  1. What is the "Effect Size" in Table 2? Is this the mean of the differences between pre- and post- values of each individual?

Author response: Dear Reviewer 2, we added a statistical explanation for Cohen’s d as an effect size or standardized mean difference measure. For a comparison between two means, Cohen's d is acceptable effect size. It can be used to complement the reporting of t-test and ANOVA results [31]. Cohen’s d represented the standardized mean difference between two groups, subtract one group's mean from the other's (M1 – M2), and divide the result by the population's standard deviation (SD) [31].

===========================

  1. Is Table 4 presenting the results from a multivariable model? It would be helpful to clarify in the table description whether those results are all part of a multivariable model.

Author response: Footnote included for the adjusting variables. Note: ANCOVA adjusting for the following factors: marital status, employment status, smoking status, and psychotropic medication and covariates age, physical activity, baseline body mass, and baseline PHQ-9.

This is also described explicitly in the methods.

===========================

  1. It is not clear whether this study was registered at a clinical trials site such as clinicaltrials.gov? Although not randomized, since it involves an intervention applied to humans, the study likely should be registered.

Author response: this was not registered in clinicaltrials.gov or WHO RCT registries due to its non-randomized nature and that fasting was a religious practice during the holy month of Ramadan and not applied in the study as an experimental intervention.

===========================

Reviewer 3 Report

Thank you for the opportunity to review the paper by Jahrami et al., exploring the effect of Ramadan fasting on depressive symptoms and metabolic parameters of males with major depressive disorder.

The aim, methods, strengths, and limitations of the study are clearly described and presented in the manuscript.

I have the following comments / suggestions:

  1. Abstract: please define the abbreviation MDD when first mentioned.
  2. Abstract: “components of the and metabolic syndrome…” There is a typo here, please correct.
  3. Abbreviations used in the tables should be better explained below each table. This would facilitate the reading and interpretation of the data presented in the tables.
  4. I feel that the presentation of the results is mainly based on the tables. The text of the results section could be more detailed and clearer. For instance, instead of simply writing that “the FG had statistically significant changes in ….”, the authors could additionally describe whether these parameters were positively or negatively affected by Ramadan.
  5. How do the authors explain the deterioration in depressive symptoms observed in the NFG (as shown in Table 2), taking into account the short follow-up period?
  6. The authors could explain in the discussion why randomization is unfeasible in studies investigating religious types of fasting (i.e. randomization cannot be performed without violating the religious beliefs / practices of the participants, please see Karras et al., Nutrients, 2021, 13, 1071).
  7. Given the non-randomized character of the study, I would suggest that the authors should tone down the definite character of the conclusions.

Author Response

Comments and Suggestions for Authors

Thank you for the opportunity to review the paper by Jahrami et al., exploring the effect of Ramadan fasting on depressive symptoms and metabolic parameters of males with major depressive disorder.

The aim, methods, strengths, and limitations of the study are clearly described and presented in the manuscript.

Author response: Dear Reviewer 3, we would like to thank you for your thoughtful comments and efforts towards improving our manuscript. All the comments have been taken into account when preparing the revision. We have addressed each and every comment. Changes in the manuscript are highlighted in bold red font for the convenience of tracking. Thank you for your nice comments.

===========================

I have the following comments / suggestions:

  1. Abstract: please define the abbreviation MDD when first mentioned.

Author response: This was defined as major depressive disorder.

===========================

  1. Abstract: “components of the and metabolic syndrome…” There is a typo here, please correct.

Author response: this was corrected to “components of the metabolic syndrome”, deleted the extra and.

===========================

  1. Abbreviations used in the tables should be better explained below each table. This would facilitate the reading and interpretation of the data presented in the tables.

Author response: abbreviations are added to Table 2, Table 3, and Table 4.

===========================

  1. I feel that the presentation of the results is mainly based on the tables. The text of the results section could be more detailed and clearer. For instance, instead of simply writing that “the FG had statistically significant changes in ….”, the authors could additionally describe whether these parameters were positively or negatively affected by Ramadan.

Author response: We refined our results section to explain the main results. However, we relied on the Tables due to a large number of presented parameters, given that abbreviations are described now below the tables. Tables are easier to read.

===========================

  1. How do the authors explain the deterioration in depressive symptoms observed in the NFG (as shown in Table 2), taking into account the short follow-up period?

Author response: Deterioration in the depressive symptoms observed in the non-fasting group can be explained by possibly the fact that previous studies in healthy people [34] and patients with diabetes [35] showed that fasting leads to improvement in stress scores, which may explain our findings.

===========================

  1. The authors could explain in the discussion why randomization is unfeasible in studies investigating religious types of fasting (i.e. randomization cannot be performed without violating the religious beliefs / practices of the participants, please see Karras et al., Nutrients, 2021, 13, 1071).

Author response: Future randomized studies are needed to eliminate the bias of self-selection of fasting as an intervention. Randomization is truly unfeasible in studies investigating religious types of fasting, as the participants are practicing their faith during fasting in the mont of Ramadan [44]; nonetheless, more rigorous research is needed.

===========================

  1. Given the non-randomized character of the study, I would suggest that the authors should tone down the definite character of the conclusions.

Author response: The tone of the conclusion was lowered; we agree with the reviewer.

===========================
